# Protective Effects of ζ-Carotene-like Compounds against Acute UVB-Induced Skin Damage

**DOI:** 10.3390/ijms241813970

**Published:** 2023-09-12

**Authors:** Liping Zhang, Shaoxin Liang, Zhi Zhang, Kai Wang, Junhan Cao, Mengke Yao, Ling Qin, Changfeng Qu, Jinlai Miao

**Affiliations:** 1Department of Special Medicine, School of Basic Medicine, Qingdao University, Qingdao 266071, China; 2020010074@qdu.edu.cn (L.Z.); 2021010157@qdu.edu.cn (Z.Z.); 2Key Laboratory of Marine Eco-Environmental Science and Technology, First Institute of Oceanography, Ministry of Natural Resources, Qingdao 266061, China; liangshaoxin@fio.org.cn (S.L.); wk2303@stu.ouc.edu.cn (K.W.); caojunhan@fio.org.cn (J.C.); ymk@stu.ouc.edu.cn (M.Y.); qinling@fio.org.cn (L.Q.); 3Qingdao Pilot National Laboratory for Marine Science and Technology, Qingdao 266237, China; 4Marine Natural Products R&D Laboratory, Qingdao Key Laboratory, Qingdao 266061, China

**Keywords:** ζ-carotene-like compounds, UVB, skin, antioxidant, ROS

## Abstract

The previous study successfully established an expression strain of ζ-carotene-like compounds (CLC) and demonstrated its remarkable antioxidant activity, which exhibited resistance to photodamage caused by UVB radiation on the skin following gavage administration. The objective of this study was to investigate the impact and mechanism of CLC on UVB-induced skin damage through topical application. Cell viability, anti-apoptotic activity, ROS scavenging ability, the inhibition of melanin synthesis, the regulation of inflammatory factors and collagen deposition were assessed in cells and mice using qRT-PCR, WB, Elisa assays, immunohistochemistry staining and biochemical kits, etc. The experimental results demonstrated that CLC-mitigated apoptosis induced by UVB irradiation up-regulated the Keap1/Nrf2/ARE antioxidant pathway to attenuate levels of ROS and inflammatory factors (NF-κB, TNF-α, IL-6 and IL-β), and suppressed MAPK/AP-1 and CAMP/PKA/CREB signaling pathways to mitigate collagen degradation, skin aging and melanin formation. In conclusion, this study underscored the potential of CLC as a safe and efficacious source of antioxidants, positioning it as a promising ingredient in the formulation of cosmetics targeting anti-aging, skin brightening and sunburn repair.

## 1. Introduction

Carotenoids are lipid-soluble pigments dominated by C_40_ interconnected polyene skeletons featuring multiple conjugated double bonds. The primary sources of these compounds are derived from photosynthetic plants, encompassing a variety of fruits, vegetables, and flowers. However, carotenoids have also been identified in non-photosynthetic organisms such as fungi (*Umbelopsis isabellina*) and bacteria (*Deinococcus-Thermus*) [1]. The color and functional activity of the compounds also vary depending on the presence of conjugated double bonds and substituents. Up to 2018, nearly 900 carotenoids had been identified, serving as active participants in photosynthesis and secondary metabolites in organisms experiencing environmental stress [2]. In addition to their vibrant hues, carotenoids serve as pigments in various industries and are highly valued for their potent antioxidant properties that aid in the prevention of diseases such as cancer, diabetes, and cataracts. As a result, they have gained widespread popularity in the pharmaceutical, food, and cosmetics sectors [3]. The sun emits three specific wavelengths of ultraviolet light, namely ultraviolet A (UVA, 320–420 nm), ultraviolet B (UVB, 275–320 nm), and ultraviolet C (UVC, 200–275 nm). Due to its short wavelength and limited penetration, UVC radiation is absorbed by the Earth’s atmosphere and does not reach the surface. As a result, it is primarily UVA and UVB rays that pose a potential risk to human health. Although UVA has strong penetrating power and can reach the dermis, UVB is currently considered more damaging to the skin than UVA [4]. With a strength 1000 times that of sunlight, UVB has become the primary wavelength responsible for inducing skin cancer [5]. While most UVB radiation is absorbed by the epidermis, its harmful impact extends beyond this layer. The UVB radiation not only induces direct DNA damage in the skin, leading to the formation of cyclobutane pyrimidine dimers (CPDs) lesions, but also initiates oxidative stress that rapidly depletes cutaneous antioxidant defenses. Moreover, keratinocytes can trigger a cascade of pro-inflammatory and intracellular signals that generate copious amounts of oxygen free radicals, resulting in skin edema, epidermal thickening, erythema, pruritus and other dermatological issues. The main strategy for preventing skin damage is therefore to employ antioxidant therapy and inhibit DNA damage.

The nuclear factor E2-related factor (Nrf2)/antioxidant response element (ARE) pathway is one of the protective mechanisms of the body in response to oxidative stress and inflammatory responses. After repeated exposure to UVB radiation, inactive Nrf2 is released from its binding with kelch-like ECH associated protein 1 (Keap1) and translocated into the nucleus, initiating the transcription of various genes related to antioxidant defense in the skin, such as heme oxygenase-1 (HO-1), NAD(P)H quinone oxidoreductase (NQO-1) and glutathione, which is considered a key and effective strategy for improving the skin’s ability to defend against light-induced damage [6,7]. Additionally, UVB-induced reactive oxygen species (ROS), including singlet oxygen, superoxide anion, hydroxyl radical, and hydrogen peroxide, can modulate the phosphorylation of protein kinases through various cascade reactions such as mitogen-activated protein kinases (MAPKs) and activated protein (AP-1) signaling pathways. This ultimately leads to an upregulation in the transcriptional expression levels of matrix metalloproteinases (MMPs), thereby accelerating skin aging [8]. Current prevention and treatment strategies for skin photoaging include oral or topical sunscreens to counteract the effects of UV radiation on DNA, cellular antioxidant balance, signal transduction pathways, immunology, and the extracellular matrix [9]. UVR (ultraviolet radiation)-induced ROS and genetic damage can also serve as crucial signals to stimulate melanogenesis and facilitate melanin transfer and deposition [10]. Therefore, antioxidants derived from plants, including polyphenols and non-polyphenols, have the potential to alleviate pigmentation or reduce melanogenesis. However, carotenoids possess multiple conjugated double bonds that not only act as antioxidants to eliminate the attack of oxygen free radicals on skin cells, but also directly affect UV absorption through their double bond structure. Therefore, carotenoids are widely utilized in the cosmetics industry due to their capacity to modulate UV-induced gene expression through various mechanisms, in addition to directly interfering with ROS or light absorption [11].

It is widely acknowledged that carotenoids cannot be synthesized by the human body and can only maintain normal physiological functions through metabolism after food intake, thus creating a significant demand for these compounds. However, the sources of carotenoids are limited, resulting in chemical synthetic products dominating the carotenoid market (80–90%), while natural sources account for a much lower proportion (10–20%) [12]. Specifically, the natural cosmetics market is one of the fastest-growing markets globally [13]. The selection of low-cost, high-efficiency and safe microorganisms for carotenoid production has garnered widespread attention. There have been numerous studies conducted on the antioxidant function and various applications of carotenoids represented by lycopene, β-carotene and astaxanthin. However, CLC—despite being an intermediate product in their synthesis—has received little attention due to difficulties in separation and low content. Nevertheless, given its multiple conjugated double bonds, the potential antioxidant function of CLC cannot be overlooked. Furthermore, the potential of CLC in preventing and treating UVB-induced skin damage remains largely unexplored. In order to broaden the functional research scope of carotenoid family members, CLC was extracted and purified from genetically engineered bacteria, which had high purity and strong antioxidant activity in vitro in previous research [14]. The previous study conducted in our laboratory demonstrated that the oral administration of CLC could effectively mitigate UVB-induced skin photodamage by modulating the composition of intestinal flora [14]. This study aimed to further investigate whether the topical application of CLC exhibited potent reparative effects on UVB-induced skin photodamage. The reparative effects of acute UVB irradiation on UVB-induced skin damage in ICR mice were investigated through histological observation, immunohistochemistry, an enzyme-linked immunosorbent assay kit (Elisa, Helsinki, Finland), and biochemical detection kits. Furthermore, the mechanisms of CLC in preventing and repairing skin damage were explored using quantitative real-time polymerase chain reaction (qRT-PCR) and Western blotting (WB). The research established a solid theoretical basis for the utilization of CLC in cosmetics or cosmeceutical products, and offered valuable insights for UVB protection and skin repair.

## 2. Results

### 2.1. Cytotoxicity and Cell Growth Curves under UVB Conditions

The CLC was incubated with epidermal cells (HaCAT) at varying concentrations for 48 h, and the results indicated that none of the concentrations were cytotoxic to cells, as depicted in Figure 1A. Under strong UVB conditions, the cells exhibited a significant decrease in viability. The treatment with 60 μg/mL CLC exhibited a gradual reduction in the decline of HaCAT viability after 48 h, and demonstrated a remarkable survival advantage at 72 h compared to the absence of CLC (Figure 1B) (*p* < 0.01). The concentration of 60 μg/mL CLC was therefore adopted for subsequent studies.

### 2.2. CLC Regulates the Keap1/Nrf2/ARE Pathway of HaCAT

Exposure to UVB radiation can induce a multitude of ROS, resulting in genetic damage, degradation, lipid peroxidation and cellular apoptosis. The Keap1/Nrf2/ARE pathway represents the quintessential antioxidant defense system, wherein Keap1 and Nrf2 typically form a complex within the cytoplasm. Upon release into the nucleus, Nrf2 binds to ARE and activates the downstream expression of key antioxidant enzymes such as catalase (CAT), peroxidase (GPX) and superoxide dismutase (SOD) that effectively scavenge peroxides and free radicals. As depicted in Figure 2A–D, the relative expression levels of Keap1 and Nrf2 were significantly upregulated and downregulated, respectively, in HaCAT cells at 4 h post UVB irradiation compared to normal cells (*p* < 0.01). Moreover, the expression levels of NQO1 and HO-1 were significantly reduced (*p* < 0.01). Conversely, under UVB irradiation, CLC-treated cells exhibited a significant decrease and increase in the relative expression levels of Keap1 and Nrf2 (*p* < 0.01, *p* < 0.05). These findings suggested that CLC activated the Keap1/Nrf2/ARE pathway. As expected, CLC treatment resulted in significantly higher expression levels of NQO1 and HO-1, both downstream targets of Nrf2, compared to UVB treatment (*p* < 0.01, *p* < 0.05). The induction of COX2 by UVB irradiation triggered an early pro-inflammatory response, promoting the conversion of arachidonic acid to prostaglandins and subsequently leading to inflammation. As demonstrated in Figure 2E–H, the exposure of HaCAT cells to UVB radiation resulted in a significant upregulation of COX2, IL-6, IL-β and TNF-α mRNA expression (*p* < 0.01). However, CLC was found to significantly inhibit their up-regulation under conditions of UVB irradiation compared to the UVB group (*p* < 0.05). In addition to the significant decrease in SOD activity, UVB treatment resulted in a significant increase in CAT and GPX activities compared to normal cells (*p* < 0.01, *p* < 0.05). Compared to the UVB control, treatment with CLC significantly increased the activities of CAT, GPX and SOD in cells (*p* < 0.05, *p* < 0.01), which effectively eliminated oxygen free radicals generated within cells and maintained REDOX homeostasis (Figure 2I–K and Appendix A). In addition, malondialdehyde (MDA) is one of the biomarkers for membrane lipid peroxidation. Compared with normal cells, UVB significantly increased the content of MDA (*p* < 0.05), indicating a significant occurrence of lipid peroxidation. CLC could significantly reduce the level of UVB-induced cell lipid peroxidation, which was consistent with the results of antioxidant enzymes (*p* < 0.05) (Figure 2L).

### 2.3. CLC Inhibits Apoptosis, CPD Formation and Aging in Cells

UVB is the primary ultraviolet radiation that triggers the apoptosis of skin cells, induces a significant production of ROS, and subsequently attacks DNA to cause its fragmentation, ultimately leading to cell apoptosis [15]. In the present study, terminal deoxynucleotidyl transferase (TdT)-mediated dUTP nick end labeling (tunel) fluorescence was employed to evaluate the degree of apoptosis in HaCAT cells following UVB irradiation, as illustrated in Appendix A. Notably, CLC exhibited a significant inhibitory effect on epidermal cell apoptosis. CPDs are the main form of UVB damage in skin DNA, which is the main factor of cell apoptosis and carcinogenesis. As illustrated in Appendix A, a substantial accumulation of CPD was observed in epidermal cells following UVB irradiation (*p* < 0.01), leading to DNA damage. However, CLC treatment significantly reduced CPD levels (*p* < 0.05), indicating its potential as an inhibitor of UVB-induced apoptosis. The senescence-associated β-galactosidase (SA-β-Gal) serves as a prototypical biomarker in senescent fibroblasts. As depicted in Figure 3, the normal human fiber cells (HS68)cells exhibited significant deformation and senescence following UVB radiation in comparison to normal cells, whereas treatment with CLC effectively mitigated cellular senescence. Therefore, it was speculated that CLC might serve as a promising agent for inhibiting UVB-induced skin aging.

### 2.4. CLC Inhibits Melanogenesis in B16-F10 Cells

Figure 4A,B show that the melanin content of mouse skin melanoma cells (B16-F10)cells increased significantly at UVB 60 mJ/cm^2^ (*p* < 0.01). The production of melanin under UVB radiation is one of the cellular self-protection mechanisms, as melanin serves as a crucial component in safeguarding the skin against ultraviolet damage [16]. However, the excessive production and accumulation of melanin can lead to pigmentation disorders, such as freckles and chloasma [17]. Treatment with 60 μg/mL CLC significantly reduced melanin content in Figure 4B (*p* < 0.01). Moreover, CLC significantly reduced the activity of TYR upon exposure to UVB at a dose of 60 mJ/cm^2^ (*p* < 0.01), which was a key enzyme in melanin synthesis (Figure 4J). That is, TYR catalyzed the formation of dopaquinone from phenylalanine and tyrosine to synthesize melanin and brown pigment. As shown in Figure 4C–I, the relative mRNA expression levels of protein kinase A (PKA), CRE-binding protein (CREB), microphthalmia transcription factor (MITF), tyrosinase (TRY), tyrosinase related protein 1 (TYRP1), tyrosinase related protein 2 (TYRP2) and ras-related protein (Rab-27A, 27A) were significantly increased (*p* < 0.05, *p* < 0.01) under the induction of UVB at 60 mJ/cm^2^ intensity, indicating that UVB at this intensity could increase the accumulation and transfer of melanin, which increased the probability of suffering from melanin accumulation leading to related diseases. However, compared with the UVB irradiation alone group, UVB with 60 μg/mL CLC treatment group could significantly reduce the relative mRNA expression of PKA, CREB, MITF, TYR, TYRP2 and Rab27A (*p* < 0.05, *p* < 0.01). In summary, CLC could reduce the production of melanin by inhibiting the CAMP/PKA/CREB signaling pathway and had high potential as a novel skin-whitening compound. However, the regulatory mechanism of CLC on CAMP is still being further studied.

### 2.5. Changes in Body Weight and Food Intake in Mice during UVB Irradiation

In this study, the body weight and food intake of mice were regularly monitored. As shown in Appendix A, there were no significant differences in body weight and food intake among the experimental treatment groups, indicating that the mice’s bodies remained stable throughout the experiment and any effects other than non-UVB radiation were excluded.

### 2.6. CLC Alleviates UVB-Induced Skin Photodamage in Mice

In this study, we simulated an intense UVB radiation experiment by exposing the skin on the back of mice. Appendix A illustrated the results after 18 days of UVB irradiation. Compared to the normal group, mice exposed to UVB showed evident erythema and scratches on their backs. Additionally, there was a highly prominent presence of inflammation-induced edema and thickening in Figure 5 (*p* < 0.01), which was significantly exacerbated upon clinical evaluation of the skin, as demonstrated in Appendix A (*p* < 0.01). In addition to mild skin edema, mice treated with CLC and vitamin E (VE) exhibited a significant improvement in skin conditions, including the disappearance of erythema (*p* < 0.01). Furthermore, the CLC and VE groups showed a notable reduction in epidermal thickness (*p* < 0.01), effectively preventing acute UVB-induced skin damage.

### 2.7. CLC Regulates the Keap1/Nrf2/ARE Pathway in Mouse Skin Tissue

When exposed to UVB radiation, the skin generates ROS, leading to the inflammation and upregulation of COX2 expression. This in turn enhances the synthesis of prostaglandin E 2 (PGE2) through a synergistic interaction with nitric oxide, which is implicated in UVB-induced erythema and melanin accumulation. Simultaneously, in vivo inflammation can activate the Keap1/Nrf2/ARE signaling pathway, thereby inhibiting COX2 expression and enhancing the antioxidant pathway, which is crucial for attenuating UVB-induced skin damage. UVB irradiation significantly upregulated the mRNA expression of COX2 (*p* < 0.05), as demonstrated in Appendix A, indicating a pronounced early inflammatory response. However, the application of CLC and VE effectively inhibited the expression of COX2 (*p* < 0.05) (Figure 6A,B). UVB-induced inflammation could activate the Keap1/Nrf2/ARE pathway, which is involved in antioxidant response. In the absence of UVB stimulation, an inactive complex of Nrf2 and Keap1 was formed in the cytoplasm. Nrf2 was activated and translocated to the nucleus in response to external stimuli, where it upregulated antioxidant factors and enzymes. As depicted in Figure 6A,C–F and Appendix A, UVB exposure significantly upregulated the protein expression of Keap1, Nrf2 and downstream NQO1 (*p* < 0.05, *p* < 0.01). However, there was no significant difference observed in mRNA expression. Notably, UVB exposure markedly raised the mRNA expression of downstream HO-1 (*p* < 0.01). Moreover, CLC treatments were observed to decrease the protein expressions of Keap1 and enhance Nrf2, as well as HO-1 and NQO1, respectively (*p* < 0.05, *p* < 0.01), as shown in Figure 6. Therefore, it could be demonstrated that the CLC further augmented the antioxidant activity of the Keap1/Nrf2/ARE pathway upon UVB stimulation. This study further investigated the mRNA expression and concentration levels of inflammatory factors, as well as the activity of antioxidant enzymes in mouse skin. The results presented in Figure 6G–J and Appendix A, and the UVB exposure significantly upregulated both the mRNA expression and concentration of inflammatory factors (*p* < 0.01, *p* < 0.05). Consistent with these findings, both CLC and VE were found to effectively reduce the expression and accumulation of inflammatory factors in the skin (*p* < 0.05, *p* < 0.01). UVB did not significantly increase the activity of CAT and GPX, with the exception of SOD. Similarly, CLC and VE did not significantly increase the activity of SOD, but significantly increased the activity of CAT and GPX (*p* < 0.05, *p* < 0.01) (Figure 6K–M). This phenomenon was consistent with a previous study that suggested that carotenoids possessed antioxidant capacity and could replace SOD in performing this function within the body [18]. The level of MDA serves as an indicator of the extent of membrane peroxidation. UVB radiation could induce significant skin lipid peroxidation and accelerate skin aging (*p* < 0.01) (Figure 6N). The CLC and VE exhibited significant efficacy in reducing MDA levels (*p* < 0.01, *p* < 0.05). These findings suggested that CLC and VE effectively protected the skin against UVB-induced damage by enhancing its antioxidant capacity. 

### 2.8. The Activation of the MAPK/AP-1 Pathway by CLC Reduces Skin Collagen Loss in Mice

It is widely recognized that AP-1 acts as a pivotal transcriptional activator for MMPs, and MAPKs play a crucial role in activating AP-1 [19]. Therefore, both qRT-PCR and WB techniques were employed in this study to quantify the expression levels of MAPK/AP-1. The results are presented in Figure 7A–K and Appendix A, demonstrating a significant upregulation of c-Jun N-terminal kinase (JNK), extracellular signal regulated kinase (ERK), P38 and AP-1 expressions upon UVB irradiation (*p* < 0.05, *p* < 0.01). Additionally, phosphorylated forms of p-JNK, p-ERK, p-c-FOS and p-Jun were also found to be increased (*p* < 0.05, *p* < 0.01), indicating the activation of the MAPK/AP-1 pathway through UVB radiation. However, CLC and VE significantly suppressed the expression and phosphorylation of MAPKs and AP-1 (*p* < 0.05, *p* < 0.01). Supplementary Figure 6E–G reveal that UVB exposure significantly upregulated the mRNA expression levels of MMP-1 and MMP-2 (*p* < 0.01), and CLC treatment not only significantly increased the mRNA expression of type I collagen but also markedly decreased the expressions of MMP-1 and MMP-2 compared to the UVB group alone (*p* < 0.01, *p* < 0.05). Therefore, it could be inferred that CLC mitigated skin collagen degradation induced by UVB damage through the inhibition of the MAPK/AP-1 signaling pathway, thereby ameliorating cutaneous aging and erythema.

The aforementioned inference could be further supported by employing immunohistochemistry and Masson staining techniques to quantitatively assess the skin aging induced by the resistance of CLC against UVB radiation. Consistent with qRT-PCR results, CLC and VE significantly suppressed MMP-2 expression while promoting collagen deposition in skin (*p* < 0.01) (Figure 8). In addition, HYP functions as the principal synthetic amino acid in collagen tissues, thereby providing an additional quantitative index for assessing collagen content. Procollagen 1 (PC-1) is the most abundant and ubiquitous fibrous collagen variant found within the body. The Elisa assay and biochemical detection kit were employed to quantify levels of PC-I and HYP in skin samples (Appendix A), which further substantiated that CLC significantly enhanced collagen accumulation within the skin, while also inhibiting aging processes (*p* < 0.05, *p* < 0.01).

It was noteworthy that the skin water content of UVB-irradiated mice exhibited a significant decrease compared to normal mice (*p* < 0.01) in Appendix A. Considering the pivotal role of skin hydration in preserving the integrity of the skin barrier and maintaining its resilience against external stimuli, it is noteworthy to mention that atopic dermatitis, psoriasis and acne exhibited a positive correlation with heightened transepidermal water loss. HA is widely recognized for its water absorption, repair and retention functions, making it a crucial component of the skin. In addition to enzymatic hydrolysis, ROS-induced fragmentation also contributes significantly to HA degradation. The levels of ROS within the skin were significantly increased upon exposure to UVB radiation, as demonstrated in Appendix A of this study (*p* < 0.01). Consequently, there was a significant reduction in HA content in the skin (*p* < 0.01), leading to a subsequent decrease in skin hydration and a severe impairment of the skin barrier function (*p* < 0.01). However, treatment with CLC and VE demonstrated a remarkable efficacy in reducing ROS levels and preventing UVB-induced transepidermal water loss (*p* < 0.01). Regrettably, VE did not exhibit any significant impact on HA level.

### 2.9. The Inhibitory Effect of CLC on CPD and Apoptosis in Mice

UVB is directly absorbed by DNA in the skin, which can cause double- and single-strand breaks in cells, resulting in 3′ hydroxyl (OH) ends. Apoptotic cell detection and quantification were carried out using the tunel method [20]. Figure 9 showed that exposure to UVB dramatically increased the levels of CPD in skin genes and caused clearly visible skin apoptosis (*p* < 0.01), which was in line with the results of cell investigations. As anticipated, CLC treatment exhibited a considerable ability to ameliorate UVB-induced apoptosis and reduce CPD content, suggesting its protective effect on skin genes and an inhibition of apoptotic processes.

### 2.10. CLC Inhibits Melanogenesis in Mice

Melanin is a crucial component in the skin’s defense against external light damage; nevertheless, excessive melanin deposition can give rise to various dermatological issues, including skin dullness, melanoma and acne. Similar to the findings observed at the cell level, exposure to UVB radiation resulted in a substantial increase in melanin deposition and a significant up-regulation of the mRNA expression of relevant synthetic genes (*p* < 0.05, *p* < 0.01). Additionally, the expressions of genes related to melanin synthesis and transport were inhibited to varying degrees by CLC and VE (*p* < 0.05, *p* < 0.01) (Appendix A). We quantified the protein levels of PKA and CREB with WB, as depicted in Figure 10. The protein expressions of PKA and CREB were markedly increased with UVB, and p-CREB levels were likewise increased (*p* < 0.01). The phosphorylated form of CREB further activated MITF expression, increasing the synthesis of melanin. CLC and VE, on the other hand, showed inhibitory effects on PKA and CREB protein expression levels, as well as a reduction in CREB phosphorylation (*p* < 0.05, *p* < 0.01). Therefore, it could be inferred that CLC potentially attenuated melanin synthesis in the skin.

## 3. Discussion

CLC was generated directly through synthetic biology for the first time in this research, and its metabolic pathway was derived from Antarctic moss. Despite numerous studies on the antioxidant function and application of carotenoids, the functional exploration of nearly a thousand carotenoids remains limited, particularly in regard to CLC. Since CLC is an intermediate product of carotenoids and a precursor of lycopene, its low content in plants makes it difficult to separate, and the CLC standard on the market is expensive, thereby limiting the function that can be explored. In this study, CLC was obtained through extraction and separation from bacteria engineered for homologous recombination. Due to its multiple conjugated double bonds, it is capable of quenching singlet oxygen, scavenging free radicals, and preventing or terminating chain reactions caused by free radicals. The skin, as the largest organ of the human body, is constantly exposed to environmental factors such as solar radiation and air pollution, which can lead to a variety of acute and chronic diseases. The deleterious effects of UVR on the skin encompass erythema (sunburn), hyperpigmentation, photocarcinogenesis and photoaging [21]. The evaluation standard for UV damage has traditionally focused on the erythema resulting from sunburn. However, recent research has revealed that carotenoids offer protection against UV-induced pigmentation and exhibit various molecular markers of oxidative stress, surpassing the mere reduction of UV-induced erythema [22]. This study discovered that CLC performed better in antioxidant activity and UV radiation damage inhibition than VE at the same dose. We inferred that this was because the genes for CLC synthesis originated from plants in the Antarctic environment, which was harsh, causing Antarctic organisms to face a survival threat far beyond terrestrial organisms. Therefore, secondary metabolites in the body may be more powerful in structure and function to adapt to long-term harsh environments [14]. In the past 40 years, over 1600 terpenoids have been identified, many of which possess unprecedented structural features and some of which are exclusive to certain plants, fungi or marine organisms [23]. Therefore, the heterologous biosynthesis of CLC with a slightly different structure from the traditionally defined ζ-carotene might eventually lead to more effective antioxidant capacity.

ROS are primarily generated through internal metabolic processes and external stimuli in organisms. However, ROS produced via normal metabolism can be regulated and some of these molecules play crucial roles, such as regulating cell division and immune function [24]. However, the unregulated production of ROS can result in oxidative stress and chronic diseases. Direct exposure to UVB radiation could interfere with organelle function and signaling pathways, leading to ROS generation within cells, which was thought to be the primary cause of radiation-induced tissue damage. As shown in Figure 11, Keap1/Nrf2/ARE was a well-known antioxidant pathway, in which Nrf2 could activate a series of drug-metabolizing enzymes such as HO-1 and NQO1 through antioxidants and electrophiles. The activation of these enzymes requires a typical DNA sequence called the ARE, which resembles the binding motif of Nrf2. The stimulation of antioxidant enzymes can augment the process of detoxification and elimination in exogenous and endogenous chemicals, thereby effectively mitigating the crisis caused by ROS [25,26]. In both cellular and animal experiments, UVB radiation was shown to partially activate the Nrf2 signaling pathway. This was due to the activation of the Nrf2/ARE pathway transcription system being one of the main mechanisms of cellular defense against oxidative stress [27]. CLC and VE could enhance the Keap1/Nrf2/ARE antioxidant pathway, significantly reducing Keap1 mRNA expression and protein levels, promoting Nrf2 and the expression of downstream antioxidant enzymes NQO1 and HO-1, increasing SOD, CAT and GPX antioxidant enzyme activity, reducing ROS and inflammatory factor accumulation (NF-κB, IL-6, IL-β and TNF-α) as well as lipid peroxidation levels in skin tissue. UVB radiation directly induced DNA damage in the form of CPDs and indirectly through ROS-mediated 8-OxoG. As a result, skin exposed to UVB underwent a significant increase in CPD content. However, CLC not only significantly reduced CPD content but also greatly mitigated the rise in the skin cell apoptosis rate. This implied that CLC modulated the activity of antioxidant enzymes by augmenting the expression of the Keap1/Nrf2/ARE signaling pathway, thereby alleviating oxidative stress, skin DNA damage and apoptosis.

The MAPK signaling pathway plays a pivotal role in skin aging caused by collagen degradation induced by MMPs, as it facilitates the transmission of extracellular signals to the nucleus, thereby activating transcription factors and inducing the expression of target genes [28]. GTP-binding proteins Rac, Ras and Cdc42 in skin surface cells are activated upon repeated exposure to UVB. Ras facilitates the activation of ERK by recruiting Raf-1 from the plasma membrane, while Rac and Cdc42 bind to MEKK1 to activate MAPK consisting of ERK, JNK and p38 [29]. AP-1 is the downstream activator of MAPK, and Jun combines with c-Fos to form heterodimer AP-1. Upon phosphorylation with c-Fos, Jun binds to target gene promoters to activate MMPs expression, in which MMP-1 and MMP-2 have been confirmed to reduce the accumulation of collagen and elastic protein [30]. To investigate the regulatory effect of CLC on the MAPK pathway, we conducted mRNA and WB experiments on key factors in the MAPK/AP-1 pathway in mouse skin. The results demonstrated that CLC exhibited significant inhibitory effects on the UVB-induced expression of ERK, JNK, P38, c-FOS, Jun, p-JNK, p-ERK, p-P38, p-c-FOS and p-Jun in mouse skin. Furthermore, the findings around MMP-1 and collagen I mRNA expression and MMP-2 immunohistochemistry, as well as HYP and procollagen levels, suggested that CLC could effectively enhance skin collagen synthesis and facilitate skin remodeling.

Current inhibitors of melanin synthesis, such as kojic acid and its derivatives (e.g., kojic acid ether derivatives), are known to inhibit TYR activity. However, these drugs exclusively target downstream signaling pathways in the melanogenesis cascade and have demonstrated limited carcinogenic potential, as well as significant adverse effects such as pigmentation alterations. Consequently, there is an imperative for novel safe inhibitors of melanin synthesis that possess the capacity to be applied in both medical and cosmetic domains [31]. The inhibition of melanogenesis induced by CLC was further tested by examining the expression of genes regulating melanogenesis, including TYR. The synthesis of melanin was positively regulated by multiple genes, such as TYRP1 and TYRP2. MITF is a transcription factor with a typical helix-loop-helix-leucine zipper structure (bHLH-Zip) and can be activated through transcription via upstream CREB promoters, playing a key role in melanin synthesis by promoting the production of TYR, TYRP1 and TYRP2 [32]. CAMP is a key physiological signal molecule regulating pigmentation in various melanin-production-related pathways [33]. The accumulation of intracellular CAMP activates PKA, which in turn promotes the expression and phosphorylation of CREB, ultimately leading to the activation of MITF expression [34]. Thus, MITF, TYR, TYRP1 and TYRP2 are often used as markers of melanogenesis. The Rab27A is involved in the transport of melanin and is used as a marker for melanin transport [35]. Additionally, the UVB-activated MAPK pathway can modulate melanin synthesis, while the CAMP/PKA pathway serves as the primary regulatory mechanism [36]. The present study demonstrated that CLC effectively suppressed the activation of the MAPK signaling pathway, leading to a significant downregulation in protein expression levels of PKA, CREB and p-CREB. Additionally, CLC treatment resulted in a marked reduction in mRNA expression levels of MITF, TYR, TYRP1, TYRP2 and 27A. These findings strongly suggested that CLC exerted its inhibitory effects on melanin formation by targeting the CAMP/PKA/CREB signaling pathway. However, more research is needed to determine the precise mechanism underlying CLC’s suppression of UVB-induced CAMP rise. Nevertheless, it is evident that CLC could indirectly impede melanin deposition induced by UVB radiation. In summary, the potential of CLC as a preventive agent for skin photoaging and whitening is highly promising. Currently, efforts are underway to analyze the intricate structure of CLC in order to identify its specific structural characteristics.

## 4. Materials and Methods

### 4.1. Reagent Materials

The protein marker (14–120 kD, 14–100 kD), CCK-8, BCA protein assay kit, superoxide dismutase (SOD), catalase (CAT), malondialdehyde (MDA), peroxidase (GPX), tyrosinase (TYR) activity detection kit, hydroxyproline (HYP) detection kit, RIPA lysis buffer, tunel apoptosis assay kit, vitamin E (VE), dimethyl sulfoxide (DMSO), penicillin-streptomycin liquid, DAPI solution (ready-to-use), Senescence-Associated β-Galactosidase (SA-β-Gal) Stain Kit and terminal deoxynucleotidyl transferase (TdT)-mediated dUTP nick end labeling (tunel) were purchased from Solarbio Co., Ltd. (Beijing, China). ROS assay kit was purchased from Wuhan Servicebio Technology Co., Ltd. (Wuhan, China). Mouse skin melanoma cells (B16-F10, BNCC100309), epidermal cells (HaCAT, BNCC339817), normal human fiber cells (HS68, BRCC-400-0091) and MEMNEAA (100×) (NEAA) were purchased from BeNa Culture Collection (Beijing, China). Fetal bovine serum protein (FBS) was bought from Hangzhou Sijiqing Bioengineering Materials Co., Ltd. (Hangzhou, China). Roswell park Memorial Institute 1640 (RPMI-1640) and Dulbecco’s modified eagle’s medium (DMEM) were bought from Thermo Fisher Scientific Inc. (Waltham, MA, USA) ROS, CPD, and IL-β Elisa kits were purchased from Shanghai Keaibo Biological Co., Ltd. (Shanghai, China). NF-κB and hyaluronic acid (HA) Elisa kits were bought from Wuhan Saipei Biotechnology Co., Ltd. (Wuhan, China), and TNF-α and IL-6 Elisa kits were purchased from Shanghai Enzyme-linked Biotechnology Co., Ltd. (Shanghai, China). The procollagen 1 (PC-1) Elisa kit was bought from Beijing Zhonghao Xinsheng Technology Co., Ltd. (Beijing, China), and Trizol and One Step SuperRT-PCR Mix kit, and TransStart^®^ Top Green qPCR SuperMix (+Dye I) were bought from TransGen Biotech Co., Ltd. (Beijing, China). COX2 (66351-1-Ig), JNK (17572-1-AP), p-JNK (80024-1-RR), ERK (11257-1-AP), p-ERK (28733-1-AP), p-P38 (28796-1-AP), PKA (55382-1-AP), CREB (12208-1-AP), p-CEEB (28792-1-AP), Nrf2 (16396-1-AP), HO-1 (10701-1-AP), NQO1 (11451-1-AP), Keap1 (10503-2-AP), GAPDH (10494-1-AP) and HRP-conjugated Affinipure Goat Anti-Rabbit IgG (H + L) (SA00001-2) were purchased from Proteintech Group, Inc. (Wuhan, China). P38 (AF6456), c-FOS (AF5354) and p-c-FOS (AF3053) were bought from Affinity Biosciences Co., Ltd. (Changzhou, China). A protease inhibitor mixture containing EDTA, 100× (AR1182) was purchased from Boster Biological Technology Co., Ltd. (Wuhan, China). Antibody eluate (G2078-100ML) was bought from Wuhan Servicebio Technology Co., Ltd. (Wuhan, China). All chemicals and solvents were analytical-grade or HPLC-grade. The CLC was derived from the previous production of this experiment (the purity of CLC was 89.88%), and the engineered bacteria were deposited in the First Institute of Oceanology, Ministry of Natural Resources (Qingdao, China) [14].

### 4.2. Cell Culture and UVB Irradiation

B16-F10 and HaCAT were cultured in RPMI-1640 and DMEM medium supplemented with 10% FBS and 1% antibiotics and incubated at 37 °C and with 5% CO_2_, respectively. HS68 was cultured in DMEM supplemented with 1% NEAA, 10% FBS and 1% antibiotics and incubated at 37 °C and with 5% CO_2_. The cells were maintained at culture densities below 1 × 10^5^ cells/mL, and the medium was changed every 2 days. The UVB intensity (60 mJ/cm^2^) was described by reference [37]. Before UV irradiation, the cells were washed and covered with a thin layer of normal saline.

### 4.3. CCK8 Assay

Cell viability was analyzed using CCK-8. In brief, the HaCAT cells were seeded at a density of 5000 cells/well in a 96-well plate for 24 h, and then pre-treated to different doses of CLC (300 µg/mL, 240 µg/mL, 180 µg/mL, 120 µg/mL, 90 µg/mL, 60 µg/mL, 30 µg/mL). Initially, CLC was dissolved in DMSO and prepared as a mother liquor with a concentration of 30 mg/mL. Subsequently, different concentrations were achieved by diluting it through the medium to maintain the DMSO concentration at 1% and incubating in an incubator with cells for 48 h. Each well was washed twice with saline solution and CCK-8 solution added. After 1–4 h of incubation, the absorbance was measured at 450 nm.

### 4.4. Antioxidant Enzyme Activities

HaCAT cells were grown to a density of 1 × 10^5^ cells/mL before being treated with medium supplemented with 60 μg/mL CLC and incubated overnight. The cells were washed twice with saline to remove any other effects before being subjected to UVB irradiation under a layer of saline solution. Following irradiation, the cells were incubated in fresh culture medium at 37 °C for 4 h and digested for the subsequent detection of enzyme activities such as SOD, CAT, GPX and MDA (Appendix A).

### 4.5. Measurement of ROS Generation and Tunel

The intracellular accumulation of ROS was detected by fluorescence microscopy using DCFH-DA, which method was referenced in kit instructions and previous research [38]. The active oxygen detection method of 6-well plate adherent cells was used. Apoptosis was detected using the tunel technique to detect DNA fragmentation of apoptotic cells. Experimental procedures and methods were referred to in the instructions on the official website (Appendix A).

### 4.6. Staining of SA-β-Gal in HS68 Cells

The staining methods of SA-β-Gal and DAPI were slightly modified according to the instructions, and the cell pretreatment was the same as in Section 4.4. The difference was that HS68 continued to have DMEM medium added with CLC after UVB irradiation for 12 h, and CLC was not added to the normal group or UVB group. The rest of the steps were the same as in the instructions (Appendix A).

### 4.7. Determination of Melanin Content and Tyrosinase Activity of B16-F10 Cells

B16-F10 cells were used to measure the concentration of melanin. The cell growth density and UVB irradiation process were the same as in Section 4.4. To obtain more obvious melanin accumulation, B16-F10 cells were incubated in an incubator at 37 °C for 12 h after UVB irradiation. The assessment of melanin content was based on prior research findings [39], and the principle was colorimetry. The TYR activity assay was performed with reference to kit instructions, and enzyme activity was determined as product yield or substrate consumption per unit time under optimal reaction conditions. 

### 4.8. Experimental Animals

Fifty ICR female mice, (20 ± 3 g), which were fed with a commercial standard laboratory diet and water ad libitum, were maintained in an air-conditioned room with temperature 20–22 °C, humidity 50–60% and a 12/12 h light-dark cycle. Following one week of adaptive feeding, the mice were randomly allocated into five groups: normal control smear group (NCS), model control smear group (MCS), high-dose CLC smear group (H-ζS), low-dose CLC smear group (L-ζS) and positive control (vitamin E) smear group (VES). The NCS received no treatment, the MCS was subjected to UVB irradiation and 0.1 mL solvent, the H-ζS received UVB irradiation and 0.1 mL CLC solution with a concentration of 600 μg/mL, the L-ζS received UVB irradiation and 0.1 mL CLC solution with a concentration of 60 μg/mL and the VES received UVB irradiation and 0.1 mL VE solution with a concentration of 600 μg/mL. The optimal concentration of CLC in cell experiments was 60 μg/mL, which was subsequently used as the effective low dose. To account for differences between monolayer keratinocytes and multilayer skin tissue, a concentration of 10 times that used in vitro (i.e., 600 μg/mL) was employed as the effective high dose [40]. The preparation method of the CLC solution remained consistent with previous studies [14]. In this study, the UVB intensity (302–310 nm) was set at 300 mJ/cm^2^ and administered after depilation with a depilatory cream [41]. The mice’s body weight and food intake were regularly measured during the experiment. They were anesthetized and sacrificed after 18 days. The exposed skin on their backs was collected for subsequent detection and photographed to determine clinical skin scores based on previous studies [42].

### 4.9. Histological Observation

The dorsal skin of ICR mice was fixed and embedded in paraffin, followed by the cutting of 5 μm sections using a rotary microtome (Leica, Wetzel, Germany). Hematoxylin-eosin (HE) staining was performed to observe epidermal hyperplasia, while Masson trichrome staining was utilized for collagen deposition analysis. The staining procedures were carried out according to their respective protocols, with minimal or no alterations.

### 4.10. Tunel Staining and MMP-2 Immunohistochemistry

The procedures of tunel staining and immunohistochemistry referred to those in the previous study and were slightly modified [43,44]. Paraffin tissue embedding, sectioning, and dehydration were performed according to standard procedures. The tissue was covered with the proteinase K working solution for repair, followed by the addition of the membrane breaking working solution to cover the tissue, etc. The tunel staining was carried out according to the procedure of the tunel kit (Appendix A). The immunohistochemistry protocol involved the following steps: sections were treated with 3% hydrogen peroxide to quench endogenous peroxidase activity, followed by blocking with 3% BSA dropwise applied to cover the tissue within the histochemistry circle. The subsequent step involved incubation with an MMP-2 antibody, followed by HRP-labeled goat anti-rabbit IgG. Finally, the freshly prepared DAB color solution was utilized to stain the tunel and MMP-2 immune groups. The positive apoptotic cells were observed to be brown-yellow in the analysis of the results.

### 4.11. Biochemical Indexes of Skin Tissue

TNF-α, NF-κB, IL-6, IL-β, HA, CPD, ROS and PC-1 levels were measured using an enzyme-linked immunosorbent assay kit according to the manufacturer’s instructions (Appendix A). HYP, TYR, MDA, SOD, GPX and CAT were detected using biochemical kits following the manufacturer’s instructions (Appendix A). The calibrations of all indexes were conducted using the gravimetric method, except for GPX and HYP, while protein concentration served as the basis for calibrating the remaining indexes. Details regarding item number and manufacturer information can be found in Section 4.1.

### 4.12. RNA Extraction and qRT-PCR

The total cellular RNA was extracted with Trizol reagent and was then reverse transcribed into cDNA using One Step SuperRT-PCR Mix kit. RNA purity (OD_260/280_) was analyzed with Nanodrop (IMPLEN, Munich, Germany). The qRT-PCR was performed with the perfect Start^®^ Green qPCR SuperMix (+Dye I) kit, and the PCR cycling conditions, and the quantitative results calculation method was based on previous studies [14]. The primer sequences of genes are shown in Appendix A.

### 4.13. Western Blotting

The method previously described was employed to extract total protein from each skin sample [45]. In brief, the RAPI lysis solution for protein extraction was supplemented with appropriate amounts of protease inhibitors and phosphoprotein inhibitors, ensuring optimal conditions for efficient extraction. After the skin was incised, it underwent lysis in the lysis solution for a duration of 30 min. Subsequently, the supernatant was centrifuged at low temperatures to determine protein concentration using the BCA kit. Following this, the protein concentration was uniformly adjusted to 4 mg/mL with the lysis solution and supplemented with 6× protein loading buffer. The proteins underwent sodium dodecyl sulfate-polyacrylamide gel electrophoresis, followed by their transfer onto a polyvinylidene difluoride (PVDF) membrane (Millipore, Burlington, MA, USA). The PVDF membrane was transferred onto ice using the wet transfer method, with a time interval of 1 min per 1 KDa molecular weight (voltage 220 V). Following the transfer, the membrane was blocked in 5% skim milk at room temperature for 2 h. Both target protein and internal reference protein were presented on the same membrane with antibody eluent. Equal protein loading was assessed using mouse GAPDH as an internal control. The blots were incubated with appropriate horseradish-peroxidase-conjugated secondary antibodies and developed with an enhanced chemiluminescence detection kit. Finally, Tanon 5200 (Shanghai Tianneng Technology Co., Ltd., Shanghai, China) was utilized for capturing images and Image J (Program, NIH, Bethesda, V1.8.0.112) was employed for gray value analysis.

### 4.14. Statistics Analyses

Data were analyzed using Prism (Graph Pad, V8.0.2) and subjected to one-way analysis of variance (ANOVA) (SPSS 23.0, Chicago, IL, USA). Overall, * or # *p* values < 0.05 were considered significant, ** or ## *p* values < 0.01 were considered markedly significant and “ns” represented no significant difference.

## 5. Conclusions

In this study, CLC exhibited significant antioxidant potential in the free radical scavenging system of mouse skin. The main demonstration was primarily achieved through topical application, which exhibited a significant improvement in UVB-induced skin damage and an elevation in levels of skin antioxidant enzymes (CAT, SOD and GPX) and alleviated lipid peroxidation. After applying CLC and exposure to UVB radiation, the expression and accumulation of inflammatory factors in the skin were significantly inhibited, while the abilities of skin remodeling were improved (reducing collagen degradation and promoting collagen synthesis). CLC could also reduce melanin formation, gene damage and cell apoptosis. The above was mainly attributed to the fact that CLC could enhance the Keap1/Nrf2/ARE antioxidant pathway and exert its antioxidant capacity, thereby improving the body’s overall antioxidant capacity. Additionally, CLC inhibited MAPK/AP-1 to reduce MMPs expression and skin aging, as well as CAMP/PKA/CREB to decrease melanin transfer and accumulation. Therefore, adding CLC into sunscreens, whitening agents or reparative cosmetics was an efficacious approach to prevent and treat skin damage caused by acute photodamage.

## Figures and Tables

**Figure 1 ijms-24-13970-f001:**
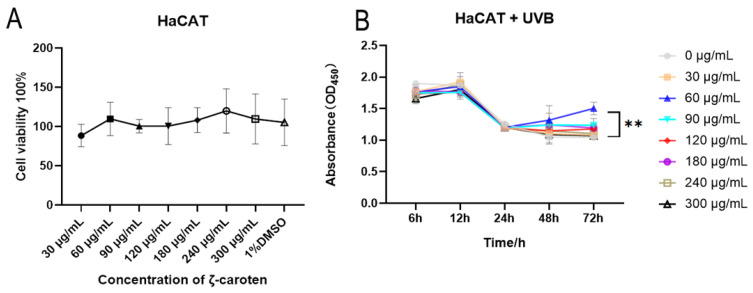
Cytotoxicity assessment of CLC (**A**) and its photoprotective effect on HaCAT cells exposed to UVB radiation (**B**). (*n* = 6, ** *p* < 0.01 vs. the 0 μg/mL CLC).

**Figure 2 ijms-24-13970-f002:**
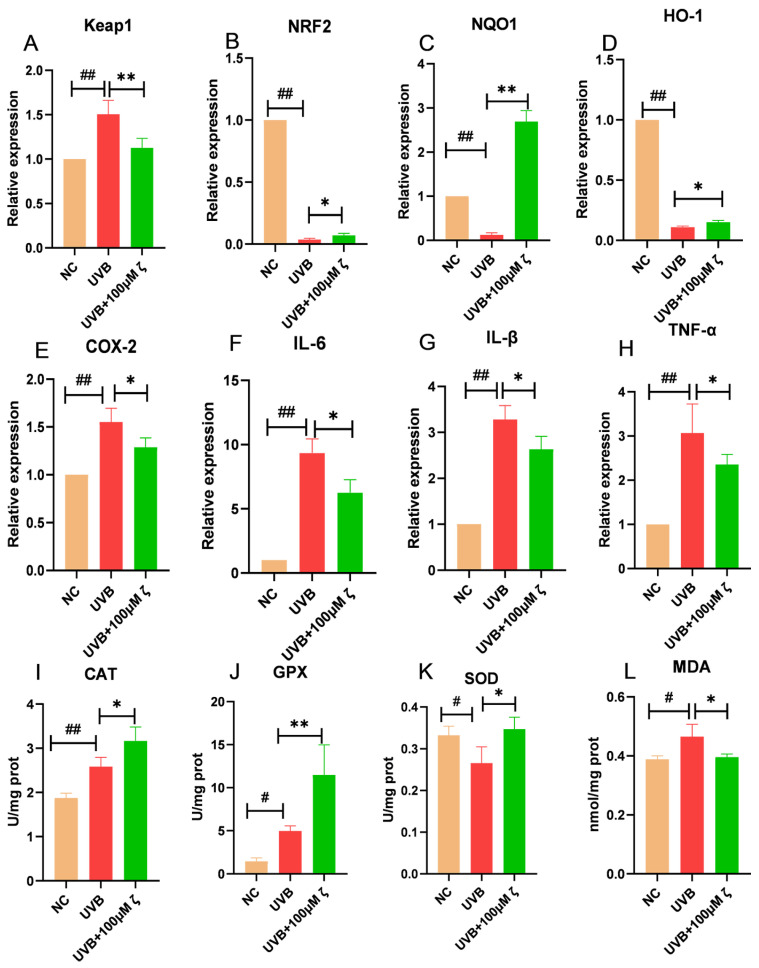
The regulatory effects of CLC on Keap1/Nrf2/ARE signaling pathway (**A**–**H**), antioxidant enzymes (**I**–**K**) and lipid peroxidation (**L**) (*n* = 3, # *p* < 0.05, ## *p* < 0.01 vs. the NC group and * *p* < 0.05, ** *p* < 0.01 vs. the UVB group).

**Figure 3 ijms-24-13970-f003:**
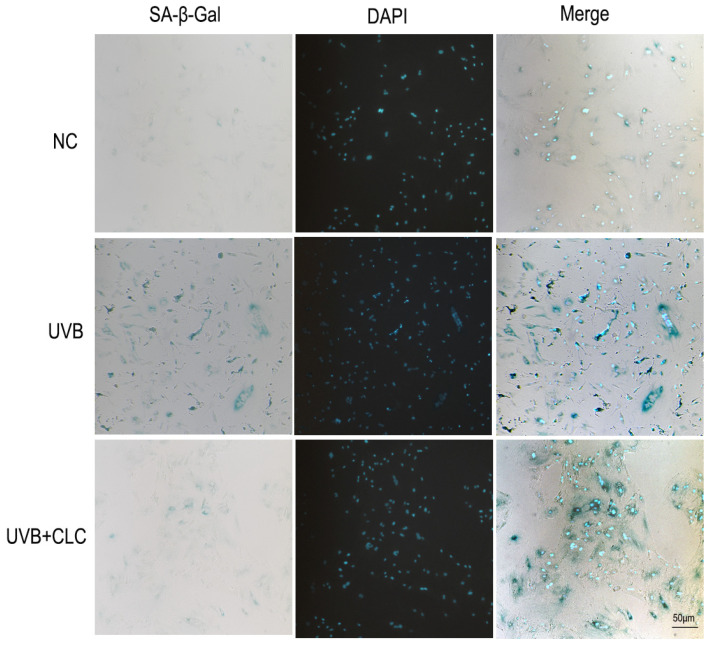
The cellular senescence was assessed through SA-β-Gal staining in HS68 (the green cytoplasm is positive for SA-β-Gal cells, and the blue fluorescence shows DAPI nuclear staining, scale bar = 50 μm).

**Figure 4 ijms-24-13970-f004:**
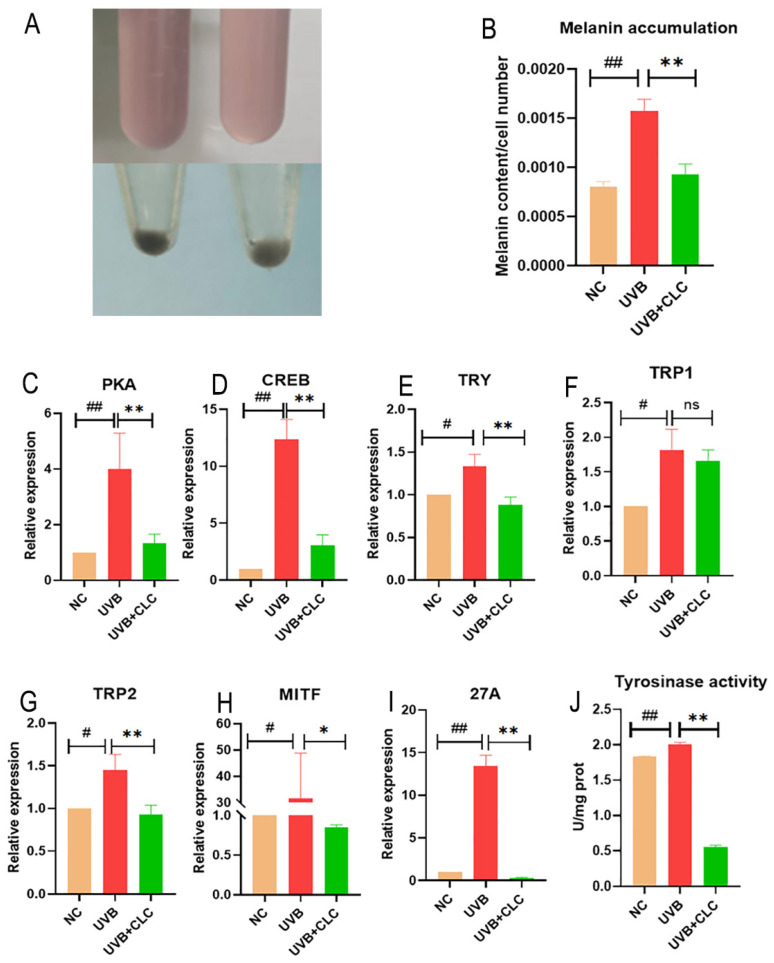
The inhibitory effect of CLC on melanin accumulation and CAMP/PKA/CREB signaling pathway under UVB treatment (on the left of (**A**) are B16-F10 cells irradiated with UVB but not incubated with CLC, while on the right are cells that are both irradiated and incubated with CLC. (**B**) is melanin content, (**C**–**I**) are the relative mRNA expression of PKA, CREB, TRY, TRP1, TRYP2, MITF and 27A, respectively, and (**J**) is tyrosinase activity. *n* = 3, # *p* < 0.05, ## *p* < 0.01 vs. the NC group and * *p* < 0.05, ** *p* < 0.01 vs. the UVB group, ns is no significance).

**Figure 5 ijms-24-13970-f005:**
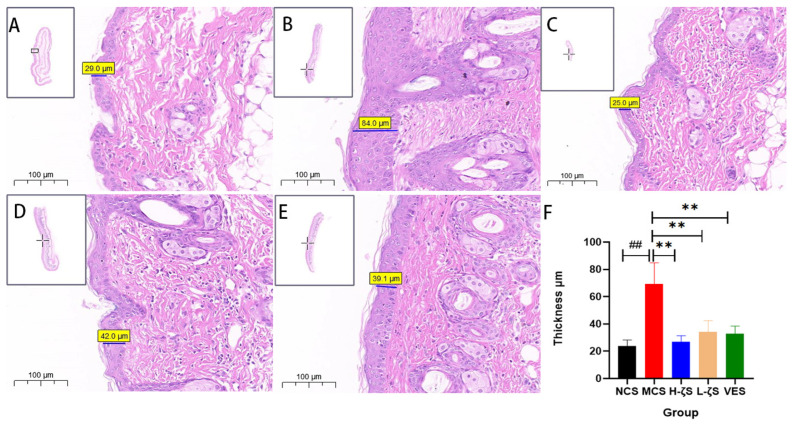
The histopathological examination results and skin thickness statistics exposed to UVB radiation. (**A**) is NCS group, (**B**) isMCS group, (**C**) is H-ζS, (**D**) is L-ζS, (**E**) is VES, (**F**) is a statistical map of epidermal thickness, scale bar = 100 μm. The figure inserted in the upper left corner is the full view of the slice, where the rectangle or cross is the specific position of the slice. ## *p* < 0.01 vs. the NCS group and ** *p* < 0.01 vs. the MCS group *n* = 3).

**Figure 6 ijms-24-13970-f006:**
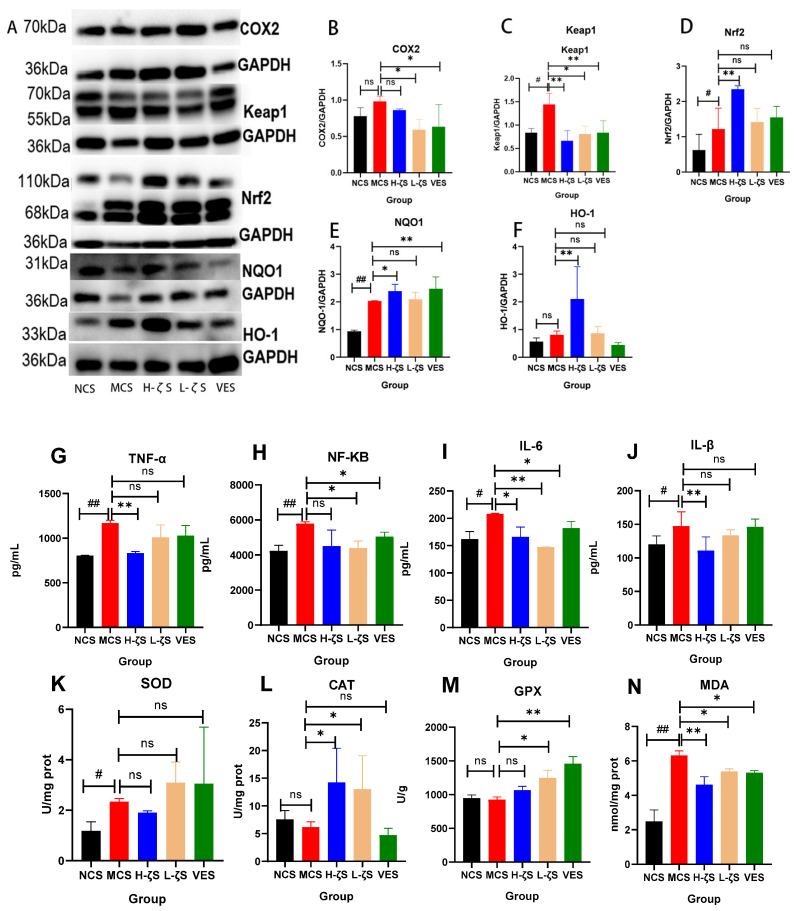
Keap1/Nrf2/ARE signaling pathway. (**A**) is the representative blots. (**B**–**F**) are the ratio of COX2, Keap1, Nrf2, NQO1 and HO-1 to GAPDH. (**G**–**J**) are inflammatory factor levels, (**K**–**M**) are antioxidant enzymes activity and (**N**) is MDA content. (*n* = 3, # *p* < 0.05, ## *p* < 0.01 vs. the NCS group and * *p* < 0.05, ** *p* < 0.01 vs. the MCS group and ns is no significance).

**Figure 7 ijms-24-13970-f007:**
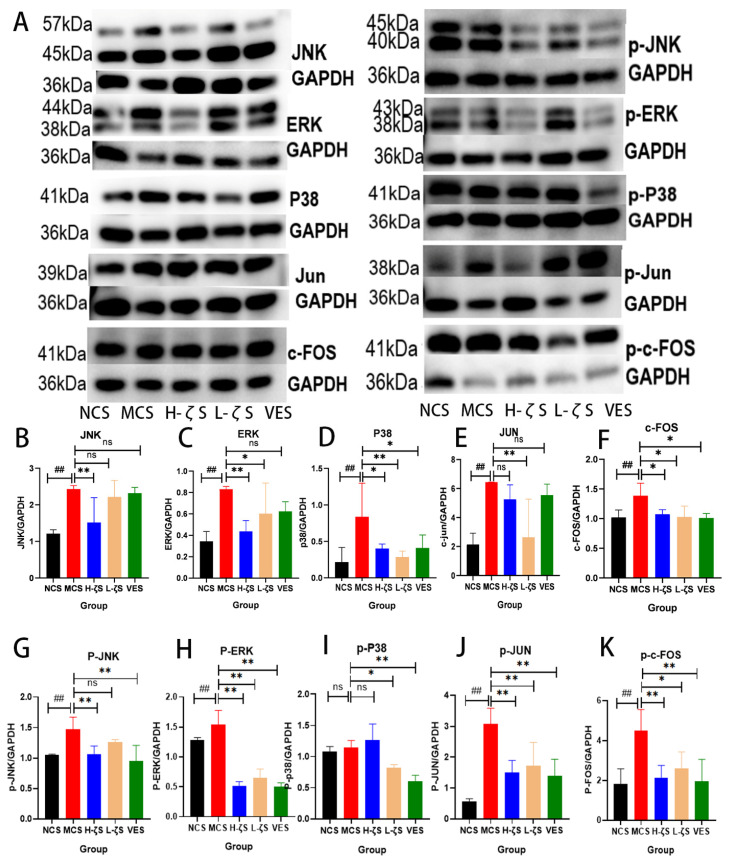
MAPK/AP-1 signaling pathway. (**A**) is the representative blots. (**B**–**K**) are the ratios of JNK, ERK, P38, Jun, c-FOS, p-JNK, p-ERK, p-P38, p-Jun and p-c-FOS to GAPDH (*n* = 3, ## *p* < 0.01 vs. the NCS group and * *p* < 0.05, ** *p* < 0.01 vs. the MCS group and ns is no significance).

**Figure 8 ijms-24-13970-f008:**
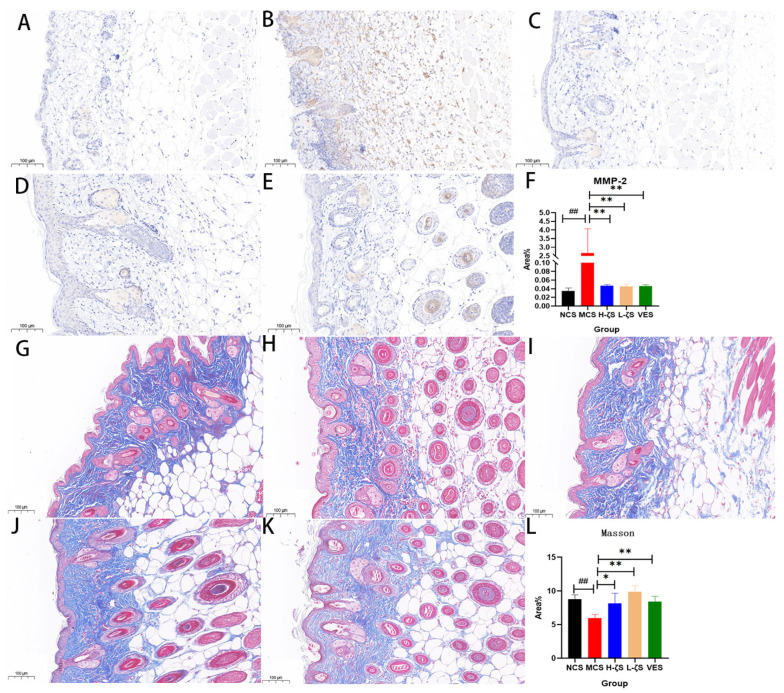
Results of MMP-2 immunohistochemistry (**A**–**F**) and Masson’s trichrome (**G**–**L**) (The collagen fibers are sky blue to light dark blue. The muscle fibers, cytoplasm, cellulose and cutin are red to purplish red. The red cells are watery red). (**F**,**L**) are relative area% of MMP-2 expression and collagen fibers, respectively ((**A**,**G**) are NCS group, (**B**,**H**) are MCS group, (**C**,**I**) are H-ζS, (**D**,**J**) are L-ζS, (**E**,**K**) are VES, *n* = 3, scale bar = 100 μm., ## *p* < 0.01 vs. the NCS group and * *p* < 0.05, ** *p* < 0.01 vs. the MCS group).

**Figure 9 ijms-24-13970-f009:**
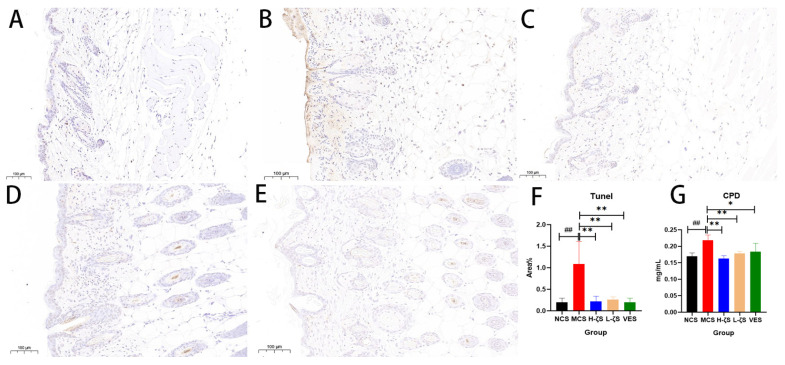
Tunel (**A**–**F**) and CPD (**G**) content in mouse skin tissue. (**F**) is the relative area% of apoptosis factor ((**A**) is NCS group, (**B**) is MCS group, (**C**) is H-ζS, (**D**) is L-ζS, and (**E**) is VES, *n* = 3, scale bar = 100 μm. ## *p* < 0.01 vs. the NCS group and * *p* < 0.05, ** *p* < 0.01 vs. the MCS group).

**Figure 10 ijms-24-13970-f010:**
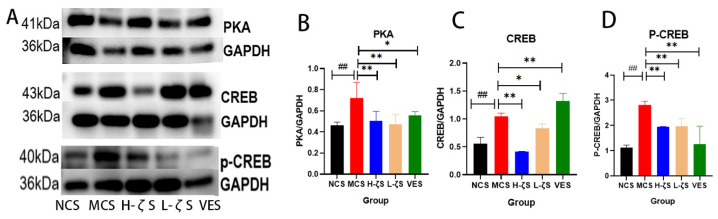
PKA/CREB signaling pathway. (**A**) is the representative blots. (**B**–**D**) are the ratios of PKA, CREB and p-CREB to GAPDH (*n* = 3 ## *p* < 0.01 vs. the NCS group and * *p* < 0.05, ** *p* < 0.01 vs. the MCS group).

**Figure 11 ijms-24-13970-f011:**
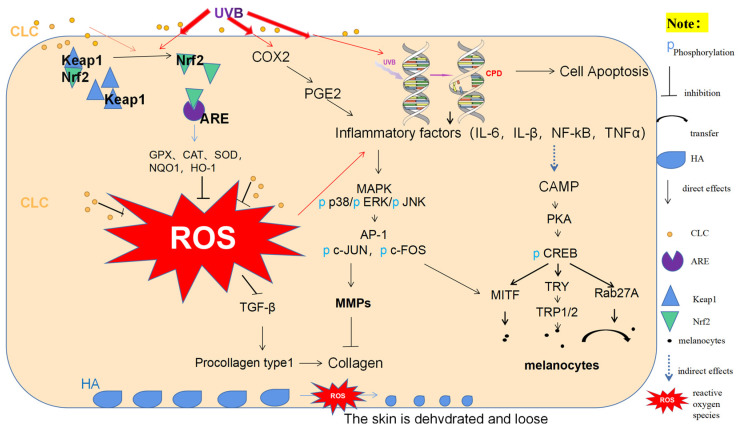
Scheme of CLC inhibition of UVB-induced photodamage.

## Data Availability

Data will be made available on request.

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
