# Peer review of "Protective Effects of ζ-Carotene-like Compounds against Acute UVB-Induced Skin Damage"

_ijms, 2023, doi:10.3390/ijms241813970_

Round 1
Reviewer 1 Report
The manuscript describes a very comprehensive study of the effects of zeta-carotene like compounds on preventing UVB-induced damage to mouse skin when applied topically. Many markers have been studied, at two different CMC doses, and compared to UVB irradiated controls and also to vitamin E. A lot of experimental work has been carried out and the results are good and show CMC to be a good candidate for addition to sunscreens, as it is protective, acts as a good antioxidant, and decreases melanin accumulation.
The main areas that I suggest should be improved are:-
1. In the introduction, I believe that it is worth stating the motivation for the study and why CMC has been chosen. Is it just that it is a natural product rather than synthetic, or are there other reasons? For example, have there been any studies comparing zeta-carotene as an antioxidant or ROS scavenger favourably against other carotenoids, such as lycopene, making it a good candidate for this study?
2. Line 90 discusses a predicted increase in personal care products by 2018. Did this increase happen? If it did, then maybe state this, or is there a new reference that could be used which predicts the increase for future years? If not, then it would be useful for the reader to state that this prediction of global demand was made in 2017 and the increase predicted for the following year.
3. I believe the Materials and Methods section needs much improvement. Firstly, the purity of CLC is given as 89.88%. It would be helpful if there is a discussion on what the main impurities are and also why these are not considered to be responsible for any of the observed effects.
4. Secondly, the methods should be expanded so that there is sufficient methodology that another researcher is able to repeat the experiments. Frequently, it is stated that kit instructions are followed (e.g. lines 165, 176 etc.). Are there references that could be included to point readers to the instructions? If not, then as a minimum, they should be written in brief, as researcers from other countries may not have access to the same kits. If this will increase this section significantly, then I suggest including them in the Supplementary Materials.
5. There is a reference with no number at the end of line 237.
6. Lines 264-6 state that a decrease in cell viability was observed for 90 ug/ml. However, the data shown in figure 1A does not seem to support this. All concentrations show the same cell viability within error, although it is 30 ug/ml (the lowest concentration tested) that seems to have the lowest viability.
7. I believe that some of the figures could be reduced, so that they have less sub figures. The ones removed can be put into the Supplementary Materials (e.g. Figure 5 A-E could be moved).
The manuscript is readable with good English, in general. However, there are several places where the past tense is used when it should be in the present and vice versa. Also, sometimes the singular is used and it should be plural. Therefore, I would recomend that this is corrected by a native English speaker, if possible, before resubmission.
Reviewer 2 Report
This paper shows the protective effect of a partially purified mixture of genetically modified bacterial extracts, termed ζ-carotene-like compound (CLC), against skin cell damage caused by UVB irradiation, both in vitro using cultured cells and in vivo using mice. If the authors claim to have observed an effect of ζ-carotene, this would be a very significant study, the first of its kind to date, and might be worthy of publication. However, CLC is a mixture containing 10.12% impurities, and we cannot be certain that all the biological effects shown here are derived from ζ-carotene-like compounds. Therefore, it is necessary to either increase the purity of the CLC a little more to clarify the stereoisomeric structure or to perform positive control experiments using commercially available ζ-carotene.
In addition, the analyses in the in vitro and in vivo experiments are different. These should be published as separate papers. In general, in vivo observations should be described as the key results, and in vitro experiments should be performed to analyze the mechanism of action.
The following is a list of some of the points I noticed in this manuscript.
1) Line 17: What do the authors mean by "the final results”?
2) Lines 22–24: If the authors are going to discuss the significance of the conjugated double bonds in ζ-carotene, they need to conduct experiments comparing phytoene, which has the fewest conjugated double bonds, with lycopene, the acyclic carotenoid with the most conjugated double bonds, and make statements based on the results.
3) Line 25: How did the authors evaluate the safety of CLC?
4) Line 59: recombinant?
5) Line 71: What does UVR stand for?
6) Line 191: What is a positive control?
7) Line 263: Presumably DMSO was used as a carrier, but the methods section should describe in detail how the CLC dissolves in the culture medium and the solubility of the CLC.
8) Lines 264–266: Figure 1A does not show a slight decrease in cell viability after exposure to 90 µg/mL.
9) Line 268: The CCK-8 assay does not measure apoptosis.
10) Line 269–270: Only one point, 60 µg/mL, significantly inhibits the UVB-induced reduction in viable cell count, while 30 and 90 show no effect at all. The authors should indicate how reproducible their observations were.
11) Line 274: It should be demonstrated whether all the effects observed in Figure 2 are dependent on the concentration of CLC.
12) Figure 2M–O: All images are blurred and it is impossible to distinguish between the three images.
13) Line 315, Supplementary Figure 1A–C: The resolution of the images is poor, all are blurred, and it is impossible to distinguish between the three images.
14) Line 336: The CLC concentration of 60 µg/ML was chosen using HaCAT cells; it is unclear if this is an appropriate concentration for B6-F10 cells.
15) Supplementary Figure 2A: What are 3.2 dates?
Round 2
Reviewer 1 Report
Thank you for responding to comments quickly. I feel all comments have now been dealt with and so recommend publication after minor changes to the English.
There are still some problems with the English, mainly with the wrong tense being used. However, these can probably be dealt with by the editorial team during publication.
Author Response
Response: Thank you very much for your approval of our revised manuscript. But I'm very sorry our grammar problems existing in the revised manuscript. The manuscript has been revised by our English-speaking colleagues and I have subsequently re-uploaded it to the system. Additionally, we have provided a list of the revision principles below:
- The present simple tense is primarily utilized to describe objective facts that are not limited by time, or to depict feelings, states, relationships, etc. that exist at the time of writing the paper. Additionally, it is customary to acknowledge previously established knowledge by citing published research in the simple present tense as a sign of respect.
- The simple past tense is employed when describing the author's own work within the paper. This includes detailing materials used, methods employed, and results obtained.
- The simple future tense is used to indicate actions or states of existence that will occur after the completion of the paper. For instance, it can be utilized to propose future research directions.
Reviewer 2 Report
I have no more comment.
Author Response
Response: Thank you very much for your approval of our revised manuscript.